

# Niraparib restrains prostate cancer cell proliferation and metastasis and tumor growth in mice by regulating the lncRNA MEG3/miR-181-5p/GATA6 pathway

Ji Cheng, Yi Sun, Huacai Zhao, Wei Ren, Dan Gao, Zhigang Wang, Wei Lv and Qingchuan Dong

Department of Urology Surgery, Shaanxi Provincial People's Hospital, Xi'an, China

Corresponding author
Qingchuan Dong,
dongqc169@163.com

## ABSTRACT

**Background.** Poly (ADP-ribose) polymerase (PARP) inhibitors (PARPi), have gained approval for treating patients with castration-resistant prostate cancer (CRPC). Maternally expressed gene 3 (MEG3), a long non-coding RNA (lncRNA), plays a role in inhibiting tumorigenesis through regulating DNA repair genes. This study aimed to investigate the association between the anti-prostate cancer (PCa) effect of niraparib, a representative PARPi, and MEG3 expression, as well as explore the downstream pathway involved.

**Methods.** The levels of MEG3, miR-181-5p, GATA binding protein 6 (GATA6) in clinical samples from PCa patients were accessed by RT-qPCR. PC3 cells were treated with niraparib, and the expression of MEG3, miR-181-5p, GATA6 expression was tested. PC3 cell proliferation, migration, and invasion were tested by CCK-8, wound healing, and Transwell assays, respectively. The bindings between miR-181-5p and MEG3/GATA6 were determined by dual-luciferase reporter gene assay. Furthermore, rescue experiments were conducted to investigate the underlying mechanism of MEG3/miR-181-5p/GATA6 axis in PCa progression. Additionally, mice were injected with PC3 cells transfected with sh-MEG3 and treated with niraparib, and the xenograft tumor growth was observed.

**Results.** MEG3 and GATA6 were upregulated and miR-181-5p was downregulated in PCa patients. Niraparib treatment substantially upregulated MEG3 and GATA6, and downregulated miR-181-5p expression in PCa cells. Niraparib effectively restrained PC3 cell proliferation, migration, and invasion. MiR-181-5p targeted to MEG3, and the inhibitory effects of MEG3 overexpression on PC3 cell proliferation and metastasis were abrogated by miR-181-5p overexpression. Moreover, GATA6 was identified as a target of miR-181-5p, and GATA6 silencing abolished the inhibitory effects of miR-181-5p inhibition on PC3 cell proliferation and metastasis. Besides, MEG3 silencing could abrogate niraparib-mediated tumor growth inhibition in mice.

**Conclusions.** Niraparib restrains prostate cancer cell proliferation and metastasis and tumor growth in mice by regulating the lncRNA MEG3/miR-181-5p/GATA6 pathway.

## INTRODUCTION

Prostate cancer (PCa) is the second most common lethal cancer and ranks second in terms of mortality in males worldwide (*Gandaglia et al., 2021*). The aging population has significantly contributed to a sharp increase in the incidence and mortality of PCa. According to the 2023 cancer statistics, PCa incidence is increasing by 3% every year, which is equivalent to 99,000 new cases annually (*Siegel et al., 2023*). Traditional treatments like survey and radiation therapy have great limitations for PCa patients, with many succumbing to the disease or developing metastasis. Surgical treatment has high risk of complications, impacting patient's quality of life and carrying a potential for recurrence and distant metastasis. Meanwhile, radiation therapy has a biochemical recurrence rate of approximately 40% and may cause side effects such as frequent urination and urgency. Moreover, it may not be effective for advanced PCa. The presence of metastatic PCa has been linked with an increased risk of mortality, contributing to 13% of all cancer-related deaths and significantly impacting patients' survival and life quality (*Bartzatt, 2020*). At present, the primary therapy for PCa is androgen deprivation (ADT) therapy, which can suppress tumor growth and delay clinical tumor progression (*Achard et al., 2022*). However, the emergence of ADT resistance in PCa patients drives the disease to the castration resistant prostate cancer (CRPC) stage (*Teo, Rathkopf & Kantoff, 2019*; *Arora & Barbieri, 2018*). Currently, continuous in-depth studies have developed various new drugs for CRPC. Nonetheless, this poses a major clinical challenge, including the selection of tailored for individual patients, the optimal combination of these new effective drugs, and an exploration of the mechanisms underlying acquired resistance (*Norz & Rausch, 2021*).

Drugs targeting poly (ADP-ribose) polymerase (PARP) to regulate cell proliferation and metastasis have gradually been applied in the clinical treatment of PCa (*Risdon et al., 2021*). PARP inhibitors (PARPi) exert their effects through synthetic lethality of homologous recombination repair gene defects, such as BRAC, inhibiting DNA damage repair and promoting apoptosis in cancer cells (*Slade, 2020*; *Li et al., 2020*). PARPi inhibit the catalytic activity of PARP1 by competitively binding to its catalytic domain, preventing the repair of single-strand, and converting them to double-strand breaks. If cancer cells have homologous recombination (HR) repair gene defects, DNA damage cannot be repaired, leading to cancer cell apoptosis (*D'Andrea, 2018*). Moreover, PARPi enhance the binding strength between PARP1 and damaged DNA, inducing PARP1 trapping, which blocks the potential DNA repair pathway and ultimately kills cancer cells (*D'Andrea, 2018*). As HR repair genes are essential for DNA repair pathways, PARPi can selectively kill cells with HR repair gene defects (*Li et al., 2020*). PARPi have been approved for the treatment of breast and ovarian cancer (*Cortesi, Rugo & Jackisch, 2021*; *Mittica et al., 2018*). Olaparib and talazoparib have already been approved by the United States Food and Drug Administration (FDA) for the treatment of BRCA-mutated breast cancer based on positive outcomes in phase 3 trials (*Cortesi, Rugo & Jackisch, 2021*). Niraparib treatment has shown significantly longer progression-free survival in patients with advanced ovarian cancer compared to placebo treatment (*González-Martín et al., 2019*). The application of PARPi has also been expanded to treat advanced PCa. Olaparib treatment has been shown to improve the overall

survival rate of metastatic CRPC (mCRPC) patients with HR repair defects by promoting DNA damage-induced cell death and suppressing tumor growth (*Teyssonneau et al., 2021*). Subsequent research has further confirmed that CRPC patients with multiple DNA HR repair gene defects can also benefit from PARPi, with a comprehensive response rate of 46.7% (*Mateo et al., 2020*). In particular, niraparib and talazoparib have shown impressive performance in phase II trials for mCRPC patients (*Flippot et al., 2022*). Furthermore, niraparib treatment has improved the objective response rate and progression-free survival in patients with biallelic BRCA1/2 alterations (*Tripathi, Balakrishna & Agarwal, 2020*). Recently, the FDA approved the fixed-dose combination of niraparib and abiraterone acetate for adult mCRPC patients with deleterious or suspected deleterious BRCA1/2 mutations (*Bischoff & Barthélémy, 2023*). Although numerous biomarkers, such as BRCA mutations and other genetic mutations related to HR, have been explored. However, there are still no gold standards for determining patients who are suitable candidates for PARPi therapy. Given the significant efficacy of niraparib in mCRPC treatment, studying its related regulatory mechanisms is of great significance. It is widely recognized that the regulatory mechanisms of PARPi mainly focus on DNA genetic variations and protein expression-mediated proliferation and apoptosis. However, it is not clear whether PARPi exerts antitumor effects through the regulation of the transcriptome level. To create more precise prognostic and therapeutic indicators and identify suitable candidates for PARPi use among the patient population, we need to consider the complex interactions among various genes and proteins in the underlying mechanisms.

Multiple abnormally expressed long non-coding RNAs (lncRNAs) play crucial roles in PCa development, and have been identified as promising therapeutic targets (*Mirzaei et al., 2022*). LncRNAs can serve as prognostic and diagnostic markers in clinical settings (*Goyal et al., 2021*). Moreover, lncRNAs regulate drug resistance and immune evasion in PCa cells (*Zhang et al., 2022*). Notably, microarray and RNA sequencing technologies have identified numerous predictive lncRNAs involved in biological pathways, including ADT therapy and PARP inhibition (*Spratt, 2019*). However, it is still unclear whether lncRNAs play a role in PARPi-mediated anti-PCa effects. Evidence suggests that the lncRNA maternally expressed gene 3 (MEG3) is downregulated in PCa tissues and cells, and overexpression of MEG3 can attenuate the abilities of PCa cell proliferation, migration, and invasion by regulating the miR-9-5p/QKI-5 axis (*Wu et al., 2019*). Another study has also shown that MEG3 overexpression inhibits the viability, clonogenicity, invasion and migration of PC3 cells, as well as the tumorigenic effects of PC3 cells in mice (*Zhou et al., 2020*). More importantly, MEG3 has been found to be involved in the regulation of certain DNA repair genes. For instance, previous studies have reported that MEG3 impedes ovarian cancer cell proliferation by promoting the expression of the DNA repair gene PTEN (*Wang et al., 2018*). Additionally, it has been shown that MEG3 inhibits bladder cancer cell progression and tumor growth by promoting PTEN expression through sponging miR-494 (*Shan et al., 2020*). MEG3 suppresses the proliferation and metastasis of gastric cancer by increasing p53 transcription and expression, which helps protect the genome by coordinating various DNA damage response mechanisms (*Wei & Wang, 2017*). Furthermore, MEG3 expression is significantly upregulated after ischemia-reperfusion,
leading to decreased intact PARP1 level and increased cleaved PARP1 level, thereby promoting cell apoptosis (*Zou et al., 2019*). These studies suggest that PARP-targeted CRPC therapies may require the activation of MEG3 to regulate DNA repair genes and exert anti-PCa effects. However, it is not clear whether PRAPi can affect the expression of MEG3 in PCa cells.

In this study, we confirmed that MEG3 was conspicuously downregulated in PCa patients and cell lines. Furthermore, we found that MEG3 was upregulated in PCa cells after PARPi (niraparib) treatment, which may be associated with PARPi-mediated anti-PCa effect. Therefore, we further investigated the downstream pathways regulated by niraparib/MEG3 in PCa. This study aimed to investigate the transcriptional regulation mechanisms related to PARPi in PCa.

## MATERIALS AND METHODS

### Clinical specimens

PCa patients ($n = 20$, average age $= 51.4 \pm 8.6$ years) were recruited from Shaanxi Provincial People's Hospital. The inclusion criteria are listed below: (a) PCa diagnosis was confirmed by pathological investigations; (b) availability of comprehensive clinical information and tissue samples for experimental use; and (c) no prior administration of anti-tumor medications or treatments. Patients with other prostatic diseases, other malignant tumors, and severe complicated diseases of heart, lung, kidney and other organs or severe infectious diseases or received any anti-tumor treatment were excluded. PCa tissues and adjacent non-tumor tissues were excised from the patients during survey. All collected tissues were immediately frozen in liquid nitrogen and stored at $-80\,^{\circ}\mathrm{C}$ until further use. All samples obtained in this study were approved by the ethics committee of Shaanxi Provincial People's Hospital and abided by the ethical guidelines of the Declaration of Helsinki, and ethics committee agreed to waive informed consent.

### Cell culture and treatment

PCa cell line PC3 (article number: CRL-3471) was obtained from ATCC (Manassas, VA, USA). Cells were maintained in RPMI-1640 containing 10% fetal bovine serum, 100 U/mL penicillin, and 100 µg/mL streptomycin under 5% $CO_2$ at $37\,^{\circ}\mathrm{C}$. Cells were used for subsequent experiments after three passages. For niraparib (1038915-58-0; MedChemExpress, Monmouth Junction, NJ, USA) treatment, PC3 cells were incubated with different final concentrations of niraparib (0, 1, 2, 4, 8 µM) for different durations (0, 30, 60, 120, 240 min).

### Cell transfection

The pcDNA-MEG3, small hairpin RNA targeting MEG3 (sh-MEG3), miR-181-5p mimic, miR-181-5p inhibitor, sh-GATA6 and their corresponding negative controls were provided by Ribobio (Guangzhou, China). For pcDNA vector construction, the pcDNA.3.1 vector (V79020; Invitrogen, USA) and the DNA fragment containing the target gene (Sangon, Shanghai, China) were double-digested with restriction endonuclease BamH I (ER0055; Thermo Scientific, Waltham, MA, USA) and Age I (ER1461; Thermo Scientific, Waltham,

MA, USA), and then the two digested products were linked with T4-DNA ligase (EL0014; Thermo Scientific, Waltham, MA, USA). The recombinant vector was transformed into *E. coli* DH5 $\alpha$ competent cells and incubated for 12 h. Monoclonal colonies were selected for culture, and positive transformants were screened. The constructed vector was verified by double digestion and sequencing analysis. They were transfected into PC3 cells with Lipofectamine 3000 regent (L3000015; Invitrogen, Carlsbad, CA, USA). Specifically, 10 $\mu$L of lipofectamine 3000 reagent was diluted with 250 $\mu$L of Opti-MEM, and 10 $\mu$L of plasmids were diluted with 250 $\mu$L of Opti-MEM. The two mixtures were incubated at room temperature for 5 min, and mixed, and incubated for an additional 20 min at room temperature. When cell confluence reached 70–80%, the transfection mixture was added to a 6-well culture plate and incubated at 37 °C in 5% $CO_2$ for 48 h. The transfection concentrations were as follows: pcDNA-MEG3 (2 $\mu$g), mimic (50 nM), inhibitor (100 nM), and shRNA (1 $\mu$g).

## RT-qPCR analysis

Total RNAs were extracted from PC3 cells using TRIzol reagent (15596026; Invitrogen, Carlsbad, CA, USA), and immediately frozen at −80 °C until use. The RNA concentration was measured using NanoDrop 2000 (2000C; Thermo Fisher, Waltham, MA, USA). Complementary DNA was synthesized from the extracted RNAs using a High Capacity cDNA Reverse Transcription Kit (4368814; Applied Biosystems, Foster City, CA, USA) with the following temperature protocol: 70 °C for 5 min, 37 °C for 5 min and 42 °C for 60 min. RT-qPCR analysis was performed using SYBR Green Master Mix (A46110; Applied Biosystems, Foster City, CA, USA) under the following reaction conditions: 95 °C for 10 min, followed by 40 cycles of 95 °C for 30 s and 60 °C for 1 min. The reaction mixture included 12.5 $\mu$L of SYBR Green PCR Mix, 1.0 $\mu$L of primer (final concentration 0.5 $\mu$M), 1 $\mu$L of cDNA, and 10.5 $\mu$L of double distilled $H_2O$. The specificity of the primers was verified by melting curve analysis, and a single peak with a melting temperature Tm above 80 °C indicated good amplification specificity. LncRNA MEG3 and GATA6 expression levels were normalized to GAPDH and miR-181-5p were normalized to U6, and calculated by the $2^{-\Delta\Delta CT}$ method. The following primer sequences were used: MEG3 (forward, 5′-AGT CCA TCG CAG ATA CTG GC-3′ and reverse, 5′-GGG AAT AGG TGC AGG GTG TC-3′), GATA6 (forward, 5′-TGC AAT GCT TGT GGA CTC TA-3′ and reverse, 5′- GTG GGG GAA GTA TTT TTG CT-3′), GAPDH (forward, 5′-CGG AGT CAA CGG ATT TGG TCG TAT-3′ and reverse, 5′-AGC CTT CTC CAT GGT GGT GAA GAC-3′), miR-181-5p (forward, 5′-GAA CAT TCA ACG CTG TCG GTG-3′ and reverse, 5′-. ATC CAG TGC AGG GTC CGA GGT A-3), and U6 (forward, 5′-CTC GCT TCG GCA GCA CA-3′ and reverse, 5′-AAC GCT TCA CGA ATT TGC GT-3′).

## Co-expression network analysis

The interactions between MEG3 and miRNAs, as well as miRNAs and mRNAs were predicted using TargetScan, miRTarBase and miRDB databases. The predicted target genes were compared with the dataset, and intersection of differentially expressed miRNAs and mRNAs was obtained to identify candidate target genes. Based on the regulatory

relationships among MEG3, miRNAs and mRNAs, the MEG3-miRNA-mRNA regulatory network was established. The co-expression network charts were visualized and analyzed using Cytoscape 3.5.1 software.

## CCK-8 assay

Cell counting kit 8 (CCK-8) assay was employed to evaluate cell proliferation. After transfection and niraparib treatment, PC3 cells suspended in RPMI-1640 medium (100 μL/well) were seeded in a 96-well plate ($5 \times 10^3$/well). Cells were then cultured for 0, 24, 48 and 72 h respectively before adding 10 μL of 100 nmol/L CCK-8 (C0040; Beyotime, Jiangsu, China) solution into the culture medium in each well. After 2 h of incubation, the 96-well plate was put into the microplate reader (iMark; Bio-Rad, Hercules, CA, USA), and the corresponding wavelength (450 nm) and measurement mode were selected. The measurement operation was performed and the absorbance value were recorded.

## Cell migration assay

After transfection and niraparib treatment, PC3 cells were seeded in a 6-well plate. On the back of the 6-well plate, uniform horizontal lines were drawn with a marker pen at approximately 0.5–1 cm intervals. At least five lines were passed through each well. The cells were then incubated in 5% $CO_2$ at 37 °C until confluence reached 60–70%. Next, the cell surface was gently scratched using a sterile micropipette tip, and the detached cells were removed through PBS flushing. Afterwards, serum-free medium was added into plates and cultured for 24 h. The scratch area was monitored under a light microscope (Nikon, Japan) at different intervals (0 and 24 h), and analyzed by ImageJ 1.8.0 software (National Institutes of Health, Bethesda, MD, USA). Briefly, the image was imported into the software, and the following steps were applied successively: "Enhance Constraint (Normalize, Saturated pixels: 0.3%)", "Smooth", "Find Edges", "Adjust (Threshold 0-20)", and "Analyze-Measure" to obtain the scratch area. The percentage of cell migration was calculated by the formula: cell migration (%) = (the scratch area at 0 h the scratch area at 24 h)/the scratch area at 0 h $\times$100%.

## Transwell invasion assay

Cell invasion was detected according to the reported method (*Pang et al., 2021*). Transwell chamber (8 μm pore size; CLS3422; Corning, NY, USA) precoated with 50 μL Matrigel (2 mg/mL) were used in Transwell invasion assay. The transfected PC3 cells suspended in FBS-free DMEM were seeded in the upper chamber, followed by addition of DMEM containing 10% FBS into the lower chamber. After 24 h culture at 37 °C, the invading cells in the lower chamber were stained by 0.1% crystal violet, and then observed under a light microscope (E100; Nikon, Toyko, Japan). The images of cells were analyzed using ImageJ 1.8.0 software (National Institutes of Health, Bethesda, MD, USA). The image was imported into the software, and the following steps were applied successively: "Image", "Type", and "8-bit" to convert the image to a grayscale image, "Edit-Invert" to convert the image background to black, "Image-Adjust-Threshold" with "B&W" selected to adjust the scroll bar and remove impurities from the background, and finally "Analyze-Analyze

Particles" to obtain the analysis result, where the count represents the number of migrated cells.

## Western blot analysis

Proteins in PC3 cells or tumor tissues were extracted using RIPA assay (Invitrogen, Carlsbad, CA, USA). RIPA lysate (200 μL/well) was added to the 6-well plates, and cells were lysed on ice for 20 min. Proteins were collected by centrifugation at 12,000 rpm for 20 min. The protein samples were mixed with SDS-PAGE loading buffer at a 4:1 ratio and then boiled at 95 °C for 5 min. Afterwards, proteins (30 μg) were separated by 10% SDS-PAGE and then transferred to a PVDF membrane (IPVH00010; Millipore, Bedford, MA, USA). After blocking with 5% skimmed milk, the membranes were incubated with primary antibodies (Abcam, Cambridge, UK) including GATA6 (1 mg/mL, 1:1000; ab175349), E-cadherin (0.294 mg/mL, 1:1000; ab40772), ICAM-1 (0.624 mg/mL, 1:1000; ab109361), CD44 (1 mg/mL, 1:1000; ab243894) and GAPDH (1 mg/mL, 1:2500; ab9485), overnight at 4 °C, and then incubated with secondary antibody (2 mg/mL, 1:2000; ab6721) at 37 °C for 2 h. Protein bands were developed with the enhanced chemiluminescence regent (RPN2236; Amersham, UK), and the gray density of bands was analyzed with ImageJ software (National Institutes of Health, Bethesda, MD, USA). The image was imported into the software, and the following steps were applied successively: "Image", "Type", and "8-bit" to convert the image to a black and white image, "Process-Subtract Background-Light background" to remove impurities from the background, "Analyze-Set Measurements" with "Area", "Mean gray value", "Min & max gray value", and "Integrated density" selected as quantitative parameters, "Analyze-Set Scale" with "pixels" set as the unit of length, "Edit-Invert" to convert the image background to a bright band, and finally, the bands were circled using the irregular circle tool and "Analyze-Measurement" was clicked to obtain the gray value of the selected area.

## Dual-luciferase reporter assay

The binding sites of miR-181-5p in MEG3 and GATA6 were searched in the Starbase v3.0 software (http://starbase.sysu.edu.cn/) in reference to the reported method (*Yang et al., 2011*). We clicked the item of miRNA target and chose miRNA-lncRNA/miRNA-mRNA, and entered miR-181-5p in the miRNA item, and all lncRNAs/mRNAs have potential binding relationship with miR-181-5p will appear. Subsequently, we searched for MEG3/GATA6 to identifu the corresponding binding sites. For dual-luciferase reporter assay, the 3′ UTR sequence of the predicted target lncRNA/mRNA was inserted into the 3′ UTR of the firefly luciferase vector. Then the constructed vector was co-transfected with miRNA into cells. If miRNA can bind to the inserted 3′ UTR sequence of lncRNA/mRNA, the translation of firefly luciferase is inhibited, resulting in a decrease in fluorescence intensity. Renilla luciferase was used as an internal reference. The relative luciferase activity was determined by calculating the ratio of fluorescence values between firefly luciferase and renilla luciferase. The MEG3-wild type (MEG3-WT; 5′-AGU GAG UAA UGG UAG UGA AUG UU-3′), MEG3 mutant type (MEG3-MUT; 5′-AGU GAG UAA UGG UAG CCC CCA AU-3′), GATA6-WT (5′-CAG CAU UUU UUA UAA UGA AUG UA-3′) and GATA6-MUT

(5′-CAG CAU UUU UUA UAA AAC CCC CA-3′) reporter vectors were constructed by Transgen Biotech (Beijing, China). The fragments of MEG3 or GATA6 containing the wild or mutated miR-181-5p binding site were synthesized and cloned into pmirGLO vector (E1330; Promega, Madison, WI, USA). Next, these plasmids were co-transfected into PC3 cells with NC mimic or miR-2113 mimic using Lipofectamine 3000 reagent for 48 h at 37 °C. The relative luciferase activity was tested with a Dual-Luciferase Reporter Assay System (Promega, Madison, WI, USA).

## Animal studies

A total of 32 healthy male BALB/c nude mice (20 $\pm$ 2 g) were provided by the experimental animal center of Xi'an Jiaotong University. Animal experiments were approved and supervised by the Animal Ethics Committee of Shaanxi Provincial People's Hospital. All methods were carried out in accordance with relevant guidelines and regulations. Mice were maintained in cages under a standard experiment environment (12 h light/dark cycle, $22-25$ °C temperature, 55–60% humidity) with free access to standard food and water. Mice were randomly divided into four groups based on the random number table method: PBS, Niraparib, Niraparib+sh-NC, Niraparib+sh-MEG3 ($n = 8$ per group). After 7 days of acclimatization, PC3 cells suspended in PBS ($1 \times 10^6$, 200 μL) were subcutaneously injected into the left flanks of mice to establish a xenograft tumor model. For niraparib treatment, niraparib (50 mg/kg) was diluted in $1 \times$PBS (200 μL) and administered intraperitoneally into mice five days per week for four weeks. The same volume of $1 \times$PBS was used as control. For Niraparib+sh-NC and Niraparib+sh-MEG3 groups, PC3 cells transfected with the negative control sh-NC or sh-CENPA were injected into mice, followed by niraparib treatment. All mice were carefully nursed after treatment. Afterwards, we measured the length and width of tumors every 7 days, and tumor volume was calculated by the formula: volume $= [\text{length} \times \text{widt}h^2]/2$. After 28 days, mice were euthanized with an intraperitoneal injection of 100 mg/kg pentobarbital sodium, and tumors were excised, imaged by a camera (Z5; Nikon, Japan), and weighed. The measurement of tumor volume and weight was conducted by 2 independent researchers who were blinded to the experimental groups.

## Immunohistochemistry assay

Tumor tissues were fixed in 10% formaldehyde, embedded in paraffin, and cut into 4 μm thick slices. Then, the slices were deparaffinized in xylene and then rehydrated with series of gradient ethanol (100%, 95%, 80%, and 70%). Slices were microwaved with sodium citrate solution (10 mM, pH 6.0) and inactivated with 3% $H_2O_2$ for 10 min. Next, slices were incubated with Ki-67 antibody (1:200; ab16667; Abcam, Cambridge, UK) or negative control anti–rabbit IgG (1:100; ab313801; Abcam, Cambridge, UK) overnight at 4 °C and then secondary antibody (anti-rabbit IgG; 1:1000; ab6721; Abcam, Cambridge, UK) for 1 h. Afterwards, the slices were stained by using a DAB kit (Beyotime, Shanghai, China) and captured images with a light microscope. The images were analyzed by ImageJ 1.8.0 software (National Institutes of Health, Bethesda, MD, USA). Briefly, the image was imported into the software and calibrated using the "Analyze-Calibrate" function with "Uncalibrate OD" selected. The "Analyze-Set Measurements" function was used to set

quantitative parameters, including "Integrated density", "Area", and "Limit to threshold". The threshold was adjusted using the "Image-Adjust-Threshold" function to select all positive signals. Finally, the "Analyze-Measure" function was used to obtain the analysis results.

### Statistical analysis

Experimental data from at least three independent experiments were presented as mean $\pm$ standard deviation (SD). The cell sample size is $N = 6$, and the animal sample size is $N = 8$. SPSS 22.0 software was used for Statistical analysis. The normal distribution of data was verified by the Shapiro–Wilk test, and the homogeneity of variances was verified by the Levene's test. Student's t test was used for comparations between two groups, and one-way analysis of variance (ANOVA) followed by Tukey-Kramer correction was used for comparations among multiple groups. Non-parametric tests (Kruskal-Wallis test/Mann–Whitney test) were used if data were not normally distributed or variances were not homogeneous. $P < 0.05$ was considered statistically significant.

## RESULTS

### The intimate relationship between LncRNA MEG3/miR-181-5p/GATA6 in PCa

First, we accessed the expression of MEG3 in tumor tissues of PCa patients using RT-qPCR. The results illustrated that MEG3 was dramatically downregulated in tumor tissues compared with non-tumor tissues (Fig. 1A). Then, we found that the MEG3/miR-181-5p/GATA binding protein 6 (GATA6) axis was intimately related in PCa by Co-expression network analysis (Fig. 1B). Additionally, our results indicated a significant upregulation of miR-181a-5p (Fig. 1C) and downregulation of GATA6 mRNA (Fig. 1D) in tumor tissues of PCa patients. As expected, MEG3 expression showed a negative correlation with miR-181-5p expression (Fig. 1E), and miR-181-5p expression showed a negative correlation with GATA6 mRNA (Fig. 1F) in our recruited PCa patients.

### Niraparib treatment upregulated MEG3 and GATA6, and downregulated miR-181-5p expression in PCa cells

PARPi exert their effects by exploiting synthetic lethality in cancer cells with defects in homologous recombination repair genes, such as BRAC, thereby inhibiting DNA damage repair and promoting apoptosis in cancer cells (*Slade, 2020*; *Li et al., 2020*). It was reported that niraparib showed impressive performance in phase II trials for mCRPC patients (*Flippot et al., 2022*). Recent studies have suggested that PARPi therapy may exert anti-PCa effects through activating MEG3 and thereby promoting PARP cleavage (*Wang et al., 2018*; *Shan et al., 2020*; *Zou et al., 2019*). Thus, we explored whether the anti-PCa effect of niraparib is related to the change of MEG3 expression. We first treated PC3 cells with different concentrations of niraparib. CCK-8 assay showed that niraparib treatment restrained PC3 cell proliferation in a dose dependent manner, and 4 μM and 8 μM niraparib had comparable inhibitory activity against PCa cell proliferation (Fig. 2A). Then, we treated PC3 cells with niraparib for 0, 30, 60, 120 min. It was observed that niraparib-mediated
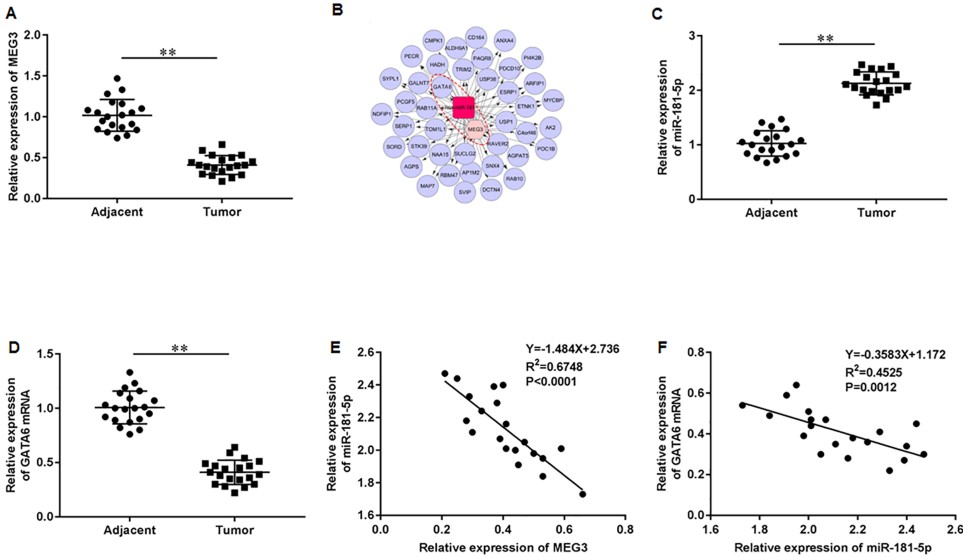

**Figure 1   LncRNA MEG/miR-181-5p/GATA6 was intimately related in PCa.** (A) MEG3 expression in tumor tissues and non-tumor tissues from PCa patients ($N = 20$) was accessed with RT-qPCR assay. (B) The MEG3/miR-181-5p/GATA6 axis was intimately related in PCa. (C) MiR-181a-5p and (D) GATA6 mRNA expression levels in tumor tissues and non-tumor tissues from PCa patients were accessed with RT-qPCR assay. (E) MEG3 expression was negatively correlated with miR- miR-181-5p expression in our recruited PCa patients. (F) MiR-181-5p expression was negatively correlated with GATA6 mRNA in our recruited PCa patients. $N = 6$. Data from at least triplicate experiments were presented as mean $\pm$ SD. $\ast\ast$ $P < 0.01$.

PC3 cell proliferation inhibition effect enhanced with incubation time (Fig. 2B). Next, we investigated the effect of niraparib on MEG3 expression in PC3 cells. RT-qPCR results revealed that niraparib treatment substantially upregulated MEG3 expression in a dose dependent manner, and which reached peak value at 4 µM (Fig. 2C). Moreover, the promoting effect of niraparib on MEG3 expression intensified with incubation time, and which reached peak value at 120 min (Fig. 2D). Besides, we also found that niraparib treatment downregulated miR-181-5p expression (Figs. 2E, 2F). and upregulated GATA6 mRNA expression (Figs. 2G, 2H). These findings suggest that the anti-PCa effects of niraparib may be mediated through the MEG3/miR-181-5p/GATA6 axis.

## Niraparib treatment restrained PCa cell proliferation, migration and invasion

We then investigate the exact effects of niraparib on Pca cell behaviors. PC3 cells were incubated with 4 nM niraparib for 120 min. Wound healing assay suggested that niraparib treatment remarkably restrained PC3 cell migration (Figs. 3A, 3B). Meanwhile, the invasion abilities of PC3 cells were suppressed by niraparib (Figs. 3C, 3D). Furthermore, it was obviously showed that niraparib incubation decreased the protein level of E-cadherin and increased the protein levels of ICAM-1 and CD44 in PC3 cells (Figs. 3E, 3F), indicating that niraparib inhibited PCa cell metastasis.

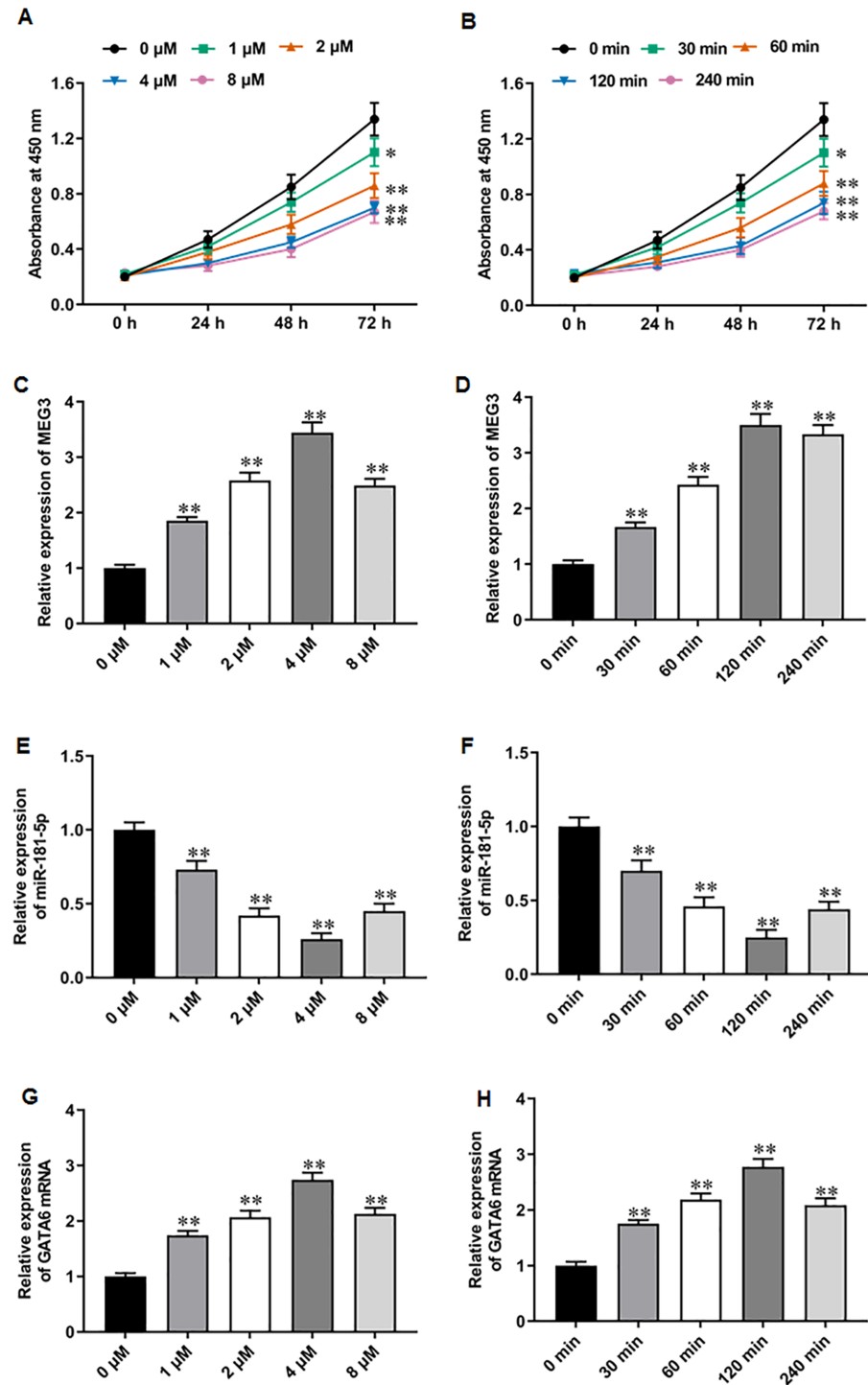

**Figure 2** **Niraparib treatment upregulated MEG3 and GATA6, and downregulated miR-181-5p expression in PCa cells.** (A) PC3 cell proliferation was accessed with CCK-8 assay after treatment with different concentrations of niraparib (0, 1, 2, 4, and 8 μM). (B) PC3 cell proliferation was accessed with CCK-8 assay after treatment with 4 μM niraparib for 0, 30, 60, 120 min. 

**Figure 2 (…continued)**
(C) MEG3 expression was accessed with RT-qPCR assay after treatment with different concentrations of niraparib (0, 1, 2, 4, and 8 μM). (D) MEG3 expression was accessed with RT-qPCR assay after treatment with 4 μM niraparib for 0, 30, 60, 120 min. (E) MiR-181-5p expression was accessed with RT-qPCR assay after treatment with different concentrations of niraparib (0, 1, 2, 4, and 8 μM). (F) MiR-181-5p expression was accessed with RT-qPCR assay after treatment with 4 μM niraparib for 0, 30, 60, 120 min. (G) GATA6 mRNA expression was accessed with RT-qPCR assay after treatment with different concentrations of niraparib (0, 1, 2, 4, and 8 μM). (H) GATA6 mRNA expression was accessed with RT-qPCR assay after treatment with 4 μM niraparib for 0, 30, 60, 120 min. $N = 6$. Data from at least triplicate experiments were presented as mean $\pm$ SD. $**P < 0.01$.

## MiR-181-5p and GATA6 are downstream genes of MEG3 in PCa cells

To elucidate the molecular mechanisms of MEG3/miR-181-5p/GATA6 axis, we used Starbase software to identify putative complementary binding sites of miR-181a-5p in the 3′-UTR of MEG3 and GATA6 (Fig. 4A). Dual-luciferase reporter assay demonstrated that miR-181-5p mimic substantially suppressed the luciferase activity of wild-type MEG3 but not mutant-type MEG3, while NC mimic had no effects on the luciferase activity of wild and mutant MEG3 (Fig. 4B). Also, the luciferase activity of wild GATA6 was obviously inhibited by transfection of miR-181-5p mimic, but the mutant GATA6 group was not affected in PC3 cells (Fig. 4C). Afterwards, we confirmed that transfection of pcDNA-MEG3 obviously facilitated MEG3 expression in PC3 cells compared with transfection of empty vector, while transfection of sh-MEG3 restrained MEG3 expression compared with transfection of sh-NC (Fig. 4D). Notably, pcDNA-MEG3 transfection remarkably inhibited miR-181-5p expression compared with empty vector, while sh-MEG3 transfection facilitated miR-181-5p expression compared with sh-NC transfection (Fig. 4E). Additionally, transfection of miR-181-5p mimic increased miR-181-5p expression compared with NC mimic, while transfection of miR-181-5p inhibitor suppressed miR-181-5p expression compared with NC inhibitor (Fig. 4F). Moreover, miR-181-5p mimic transfection prominently reduced GATA6 mRNA and protein levels compared with NC mimic, but they were elevated after miR-181-5p inhibition while transfection of miR-181-5p inhibitor elevated GATA6 mRNA and protein levels compared with NC inhibitor (Figs. 4G–4I). These results confirm that the miR-181-5p and GATA6 are downstream genes of MEG3 in PCa cells.

## MiR-181-5p overexpression reversed MEG3 overexpression-mediated inhibition of PCa cell progression

We then adopted rescue experiments to determine the roles of MEG3/miR-181-5p/GATA6 axis in PCa cell progression. PC3 cells were co-transfected with pcDNA-MEG3 and miR-181-5p mimic. First, we observed that MEG3 overexpression suppressed miR-181-5p expression in PC3 cells, whereas miR-181-5p mimic transfection increased miR-181-5p level (Fig. 5A). Then, MEG3 overexpression prominently restrained PC3 cell proliferation, which were abolished by miR-181-5p overexpression (Fig. 5B). Furthermore, MEG3 overexpression suppressed PC3 cell migration (Figs. 5C, 5D) and invasion (Figs. 5E, 5F), whereas miR-181-5p overexpression retarded these effects. Besides, MEG3 overexpression decreased the protein level of E-cadherin and increased the protein levels of ICAM-1 and CD44 in PC3 cells (Figs. 3G, 3H), while this expression pattern was reversed by miR-181-5p

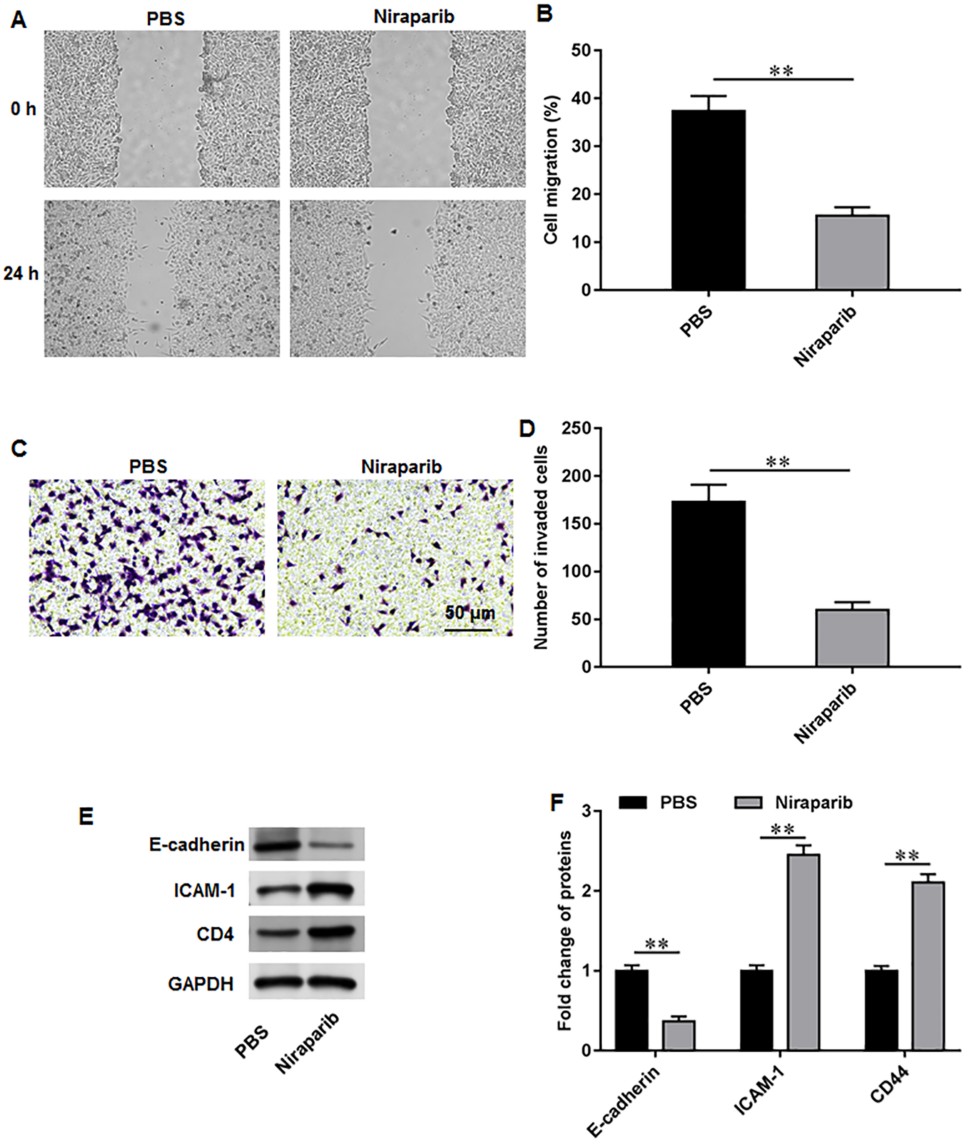

**Figure 3** **Niraparib treatment restrained PCa cell proliferation, migration and invasion.** PC3 cells were incubated with 4 nM niraparib for 120 min. (A, B) PC3 cell migration was accessed with wound healing assay. (C, D) PC3 cell invasion was tested with Transwell assay. (E, F) E-cadherin, ICAM-1, and CD44 protein levels were gauged with Western blot assay. $N = 6$. Data from at least triplicate experiments were presented as mean $\pm$ SD. $**P < 0.01$.

overexpression. These results demonstrate that MEG3 overexpression mediates PCa cell biological functions by regulating miR-181-5p expression.

## GATA6 silencing abrogated the effects of miR-181-5p inhibition on T24/DDP cell behaviors

Next, PC3 cells were transfected with miR-181-5p inhibitor and si-GATA6. Western blot results proposed that miR-181-5p inhibition markedly enhanced GATA6 expression, while si-GATA6 transfection decreased GATA6 expression (Fig. 6A). MiR-181-5p inhibition

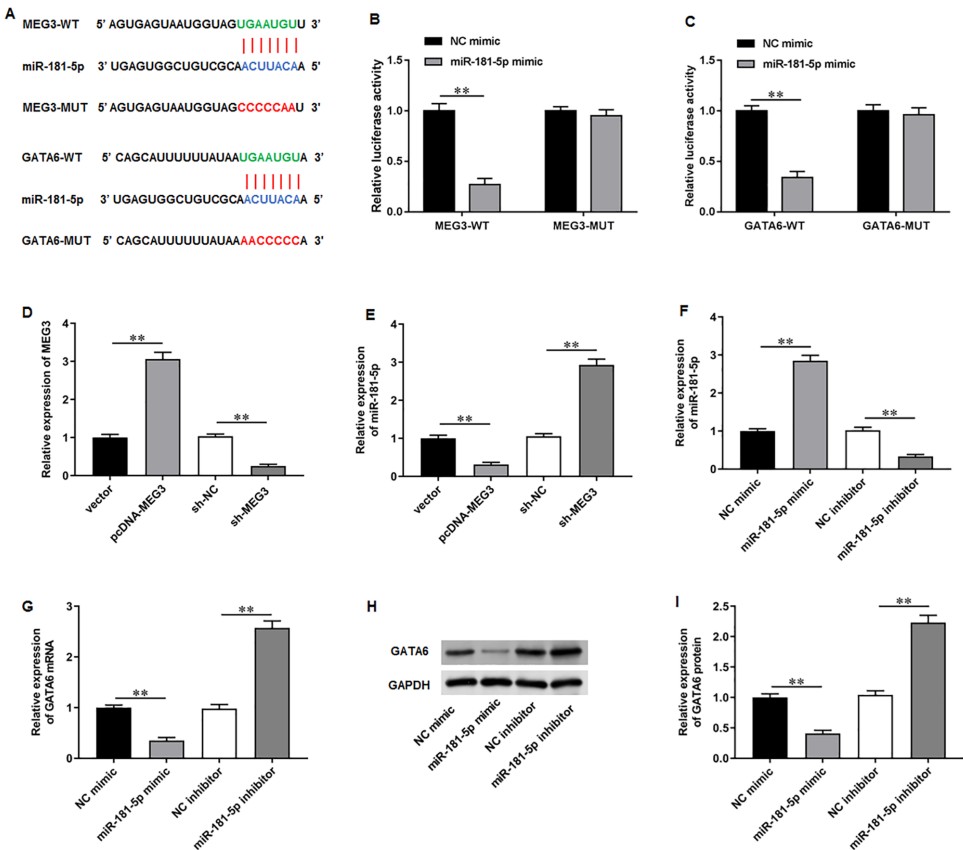

**Figure 4** **MiR-181-5p and GATA6 were downstream genes of MEG3 in PCa cells.** (A) StarBase software showed that miR-181a-5p had putative complementary binding sites with the 3′-UTR of MEG3 and 3′-UTR of GATA6. (B, C) Dual-luciferase reporter assay was applied to validate the binding between MEG3 and miR-181-5p, as well as miR-181-5p and GATA6. (D) MEG3 and (E) miR-181-5p expression was gauged with RT-qPCR assay after transfection of pcDNA-MEG3 or si-MEG3 in PC3 cells. (F) MiR-181-5p expression was gauged with RT-qPCR after transfection of miR-181-5p mimic or si- miR-181-5p inhibitor in PC3 cells. (G–I) The mRNA and protein expression of GATA6 was gauged with RT-qPCR or Western blot assay after transfection of miR-181-5p mimic or miR-181-5p inhibitor in PC3 cells. $N = 6$. Data from at least triplicate experiments were presented as mean ± SD. **$P < 0.01$.

suppressed PC3 cell proliferation (Fig. 6B), while GATA6 silencing retarded this effect. Also, miR-181-5p inhibition mitigated PC3 cell migration (Figs. 6C, 6D) and invasion (Figs. 6E, 6F), while these effects were abrogated by miR-181-5p inhibition. Additionally, our results suggested that E-cadherin level was reduced and ICAM-1 and CD44 levels were increased after miR-181-5p inhibition, while these effects were reversed by GATA6 silencing (Figs. 7G, 7H). The rescue experiment results suggest that MEG3 attenuates PCa cell progression through the miR-181-5p/GATA6 axis.

## Niraparib mitigated PCa tumor growth *in vivo* through regulating the MEG3/miR-181-5p/GATA6 axis

We finally investigated the correction between niraparib and the MEG3/miR-181-5p/GATA6 axis *in vivo*. PC3 cells were injected into mice to establish a xenograft tumor
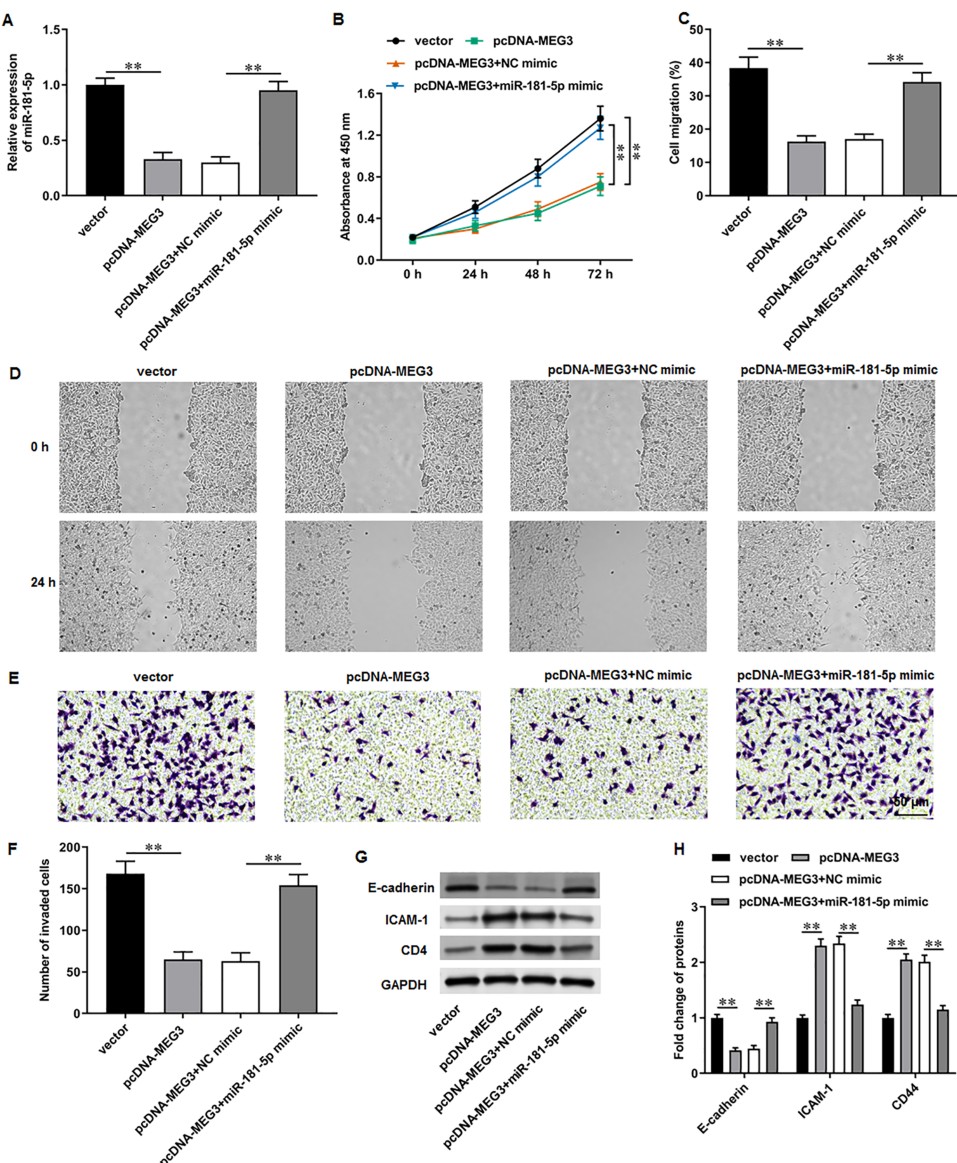

**Figure 5** **MiR-181-5p overexpression reversed MEG3 overexpression-mediated inhibition of PCa cell progression.** PC3 cells were co-transfected with pcDNA-MEG3 and miR-181-5p mimic. (A) MiR-181-5p level was tested with RT-qPCR analysis. (B) PC3 cell proliferation was accessed with CCK-8 assay. (C, D) PC3 cell migration was accessed with wound healing assay. (E, F) PC3 cell invasion was tested with Transwell assay. (G, H) E-cadherin, ICAM-1, and CD44 protein levels were gauged with Western blot assay. $N = 6$. Data from at least triplicate experiments were presented as mean ± SD. ∗∗$P < 0.01$.

model. It was clearly observed that tumor volume and weight were conspicuously decreased after niraparib injection compared with injection of PBS, whereas MEG3 silencing could retarded niraparib-mediated tumor inhibition (Figs. 7A–7C). Next, immunohistochemistry assay suggested that niraparib treatment intensified MEG3 and GATA6 expression and decreased miR-181-5p expression in tumor tissues, while MEG3 silencing abolished these effects (Figs. 7D–7F). In addition, immunohistochemistry assay illustrated that niraparib

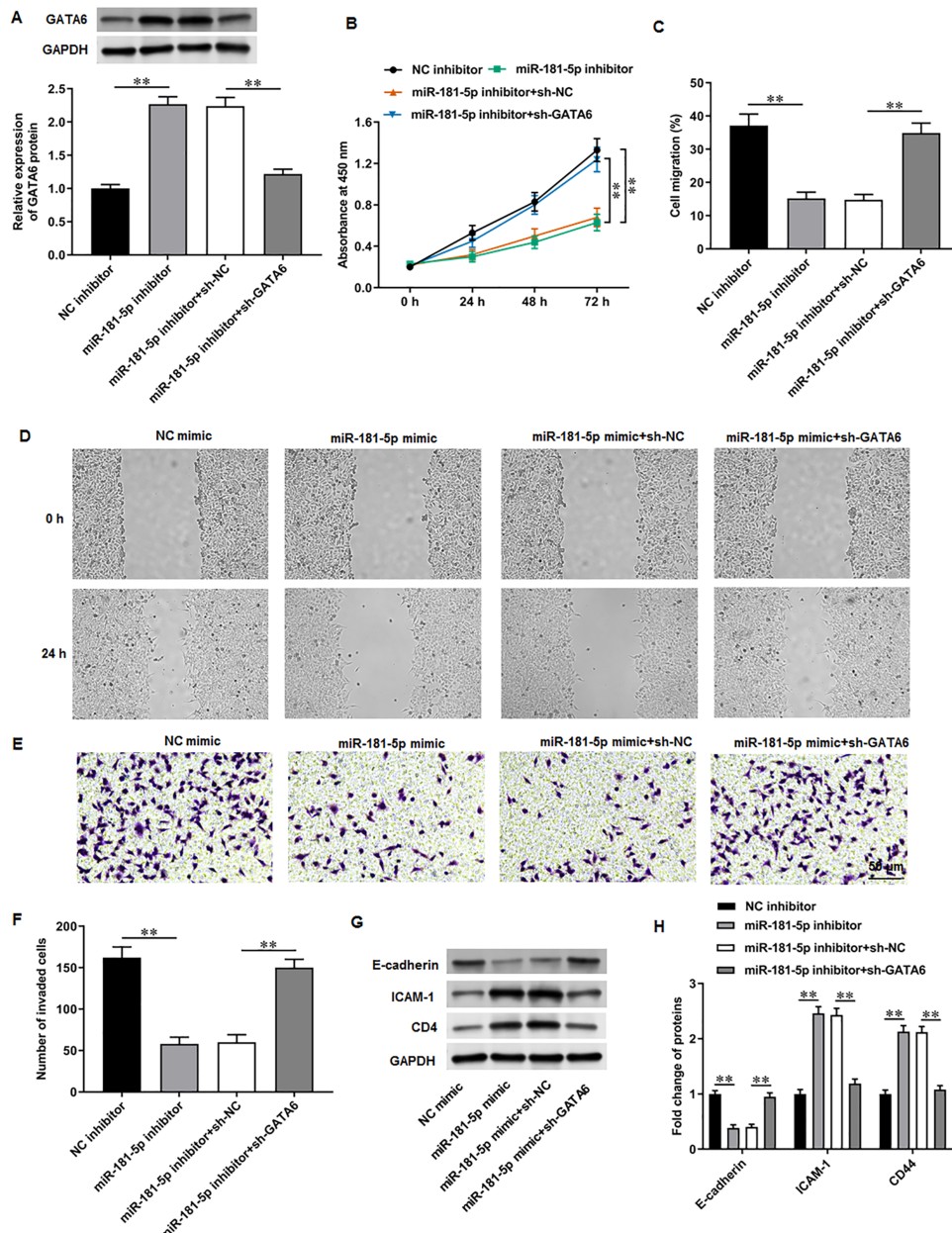

**Figure 6** **GATA6 silencing abrogated the effects of miR-181-5p inhibition on T24/DDP cell behaviors.**
PC3 cells were transfected with miR-181-5p inhibitor and si-GATA6. (A) GATA6 level was tested with
Western blot analysis. (B) PC3 cell proliferation was accessed with CCK-8 assay. (C, D) PC3 cell migration
was accessed with wound healing assay. (E, F) PC3 cell invasion was tested with Transwell assay. (G, H)
E-cadherin, ICAM-1, and CD44 protein levels were gauged with Western blot assay. $N = 6$. Data from at
least triplicate experiments were presented as mean ± SD. $**P < 0.01$.

injection reduced Ki67 protein level in tumors, which were then reversed by MEG3 silencing
(Figs. 7G, 7H). Therefore, our results proposed that niraparib mitigated PCa tumor growth
*in vivo* through regulating the MEG3/miR-181-5p/GATA6 axis.

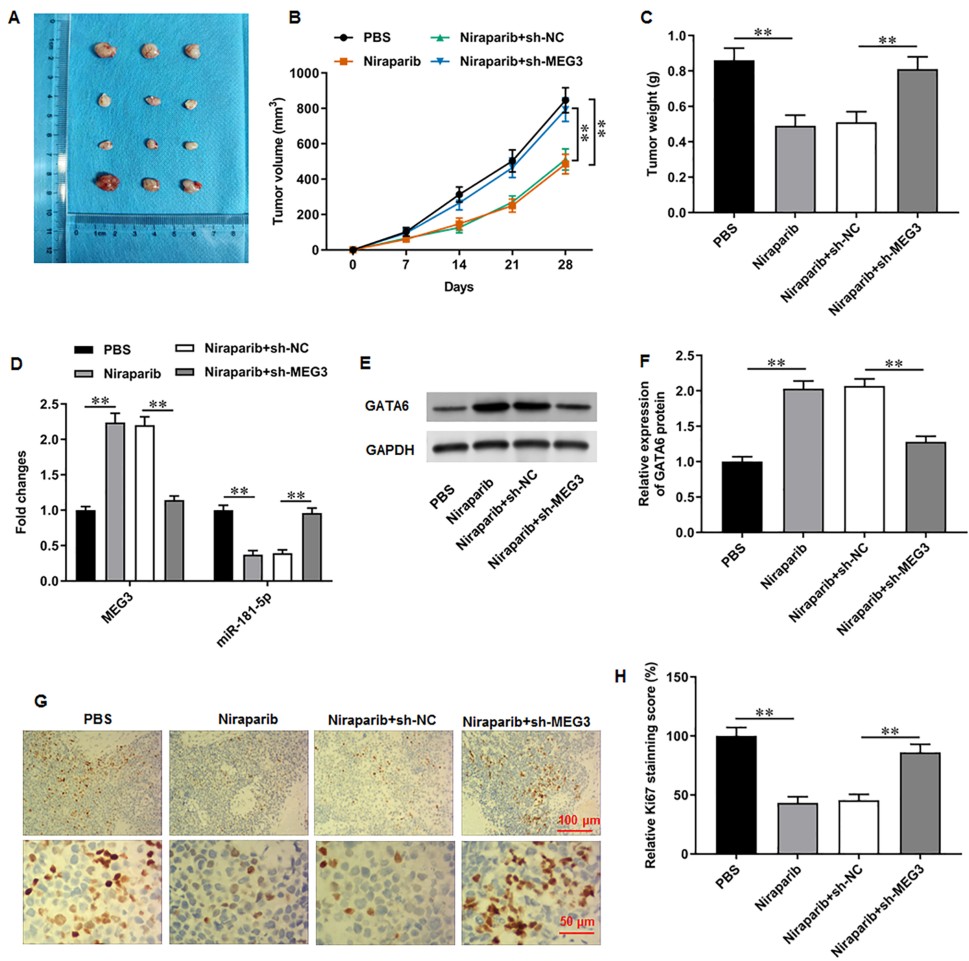

**Figure 7 Niraparib mitigated PCa tumor growth *in vivo* through regulating the MEG3/miR-181-5p/GATA6 axis.** A xenograft PCa tumor mouse model was establishd, and mice were divided into four groups: PBS, Niraparib, Niraparib+sh-NC, Niraparib+sh-MEG3 ($n = 8$ per group). (A–C) Tumor volume and weight were accessed. (D) MEG3 and miR-181-5p expression were gauged with RT-qPCR assay. (E, F) GATA6 protein level was gauged with Western blot assay. (G, H) Ki67 level was gauged with immunohistochemistry assay. $N = 8$. Data from at least triplicate experiments were presented as mean ± SD. *$P < 0.01$.

# DISCUSSION

The development of PARPi therapy has prominently improved the treatment outcomes of metastatic PCa patients with certain genetic mutations (*Grewal, Grewal & Tabbara, 2021*). It was reported that niraparib and talazoparib showed impressive performance in phase II trials for mCRPC patients (*Flippot et al., 2022*). A phase 2 clinical trial demonstrated that niraparib is relatively safe and exhibits anti-tumor activity in patients with mCRPC (*Smith et al., 2022*). Moreover, a recent study has illustrated that niraparib offers better tissue exposure and more potent tumor growth suppression in PCa bone metastasis mice compared to other PARPi (*Snyder et al., 2022*). In our study, we investigated the molecular mechanisms underlying the anti-PCa effect of niraparib.

Current evidence suggests that lncRNA MEG3 is downregulated in PCa tissues. MEG3 overexpression has been shown to mitigate PCa cell proliferation and metastasis, induce apoptosis, and attenuate tumor development in mice (*Wu et al., 2019*; *Zhou et al., 2020*). Notably, MEG3 has been found to be involved in the progression of multiple cancers through regulating some DNA repair genes, such as PTEN (*Wang et al., 2018*; *Shan et al., 2020*) and p53 (*Wei & Wang, 2017*). Importantly, it has also been found that MEG3 overexpression decreases intact PARP level and increases cleaved PARP level, thus promoting cell apoptosis (*Zou et al., 2019*). Based on these findings, we hypothesized that PARPi therapy may require the activation of MEG3 to regulate DNA repair gene and exert anti-PCA effects. The effect of PRAPi on MEG3 expression has not been studied to date. Therefore, to explore more targets for PRAPi therapy, it is of great significance to investigate the impact of MEG3 expression on PARP1 targeted CRPC treatment. As expected, our results showed that niraparib treatment upregulated MEG3 expression in PCa cells. Additionally, niraparib administration restrained tumor growth in a PCa xenograft mouse model, while MEG3 silencing treatment retarded these effects. Thus, niraparib mediated-MEG3 upregulation is a crucial mechanism for tumor inhibition.

Our study identified miR-181-5p as a highly expressed miRNA in PCa that is negatively correlated with MEG3 expression. A previous miRNA-microarray analysis identified that miR-181-5p was associated with drug resistance and efflux, and epithelial to mesenchymal transition in PCa (*Verma et al., 2019*). It has also been shown that miR-181-5p can lead to cisplatin resistance in PCa cells through complementary interactions with the 3′ UTR of the proapoptotic protein BAX transcript (*Cai et al., 2017*). Moreover, MiR-181 has been found to facilitate PCa cell proliferation and tumor development in mice through regulation of DAX1, an androgen receptor negative regulator (*Tong et al., 2014*). These findings indicate that miR-181-5p is closely related to the natural course, drug resistance, and androgen receptor resistance of PCa. Our study implied that miR-181a-5p was obviously upregulated in PCa patients, and its expression was negatively correlated with MEG3 expression. Subsequently, we confirmed miR-181a-5p as a target of MEG3 in PCa cells through Starbase database prediction and dual-luciferase reporter assay validation. Rescue experiments implicated that miR-181a-5p overexpression reversed MEG3 overexpression-mediated suppression of PCa cell proliferation and metastasis, implying that MEG3 exerted anti-PCa effect through reducing miR-181-5p expression.

Our study found that an intimate relationship between the MEG3/miR-181-5p/GATA6 axis in PCa. GATA6 is a member of the gene family with the promoter GATA core conserved sequence. An RNA-sequence analysis of tumor tissue samples from PCa patients revealed that GAGT6 is a downregulated gene in PCa (*Nikitina et al., 2017*). Moreover, lncRNA LINC00261 has been shown to intensify GATA6-mediated transcriptional inhibition and suppress PCa tumorigenesis (*Li, Li & Wei, 2020*). GATA6 has been identified as a downstream of the Linc00518/miR-216b-5p axis, and is closely related to paclitaxel resistance in PCa (*He et al., 2019*). Our further study confirmed that GATA6 mRNA was downregulated in PCa patients. GATA6 was a target gene of miR-181-5p, and its expression was suppressed by miR-181-5p. Furthermore, miR-181-5p inhibition restrained PCa cell proliferation, migration, and invasion, whereas these effects were abrogated by GATA6

silencing. Therefore, we proposed that MEG3 participated in PCa progression through the miR-181-5p/GATA6 pathway.

## CONCLUSIONS

PARPi have been approved for the treatment of PCa patients in the CRPC stage. Our study illustrated that the representative PRAPi drug niraparib, restrained PCa cell proliferation, migration and invasion and delayed tumor growth in mice by regulating the MEG3/miR-181-5p/GATA6 pathway. These findings reveal a novel molecular mechanism by which niraparib exerts its anti-PCa effects.

### Funding

This work was supported by Shaanxi Natural Science Basic Research Program "Research of the invasion and metastasis mechanism mediated by LncRNA MEG3/miR-181/GATA6 axis in prostate cancer cells" (No.2023-JC-YB-796). The funders had no role in study design, data collection and analysis, decision to publish, or preparation of the manuscript.

### Grant Disclosures

The following grant information was disclosed by the authors:
Shaanxi Natural Science Basic Research Program "Research of the invasion and metastasis mechanism mediated by LncRNA MEG3/miR-181/GATA6 axis in prostate cancer cells": No.2023-JC-YB-796.

### Competing Interests

The authors declare there are no competing interests.

### Author Contributions

- Ji Cheng conceived and designed the experiments, analyzed the data, authored or reviewed drafts of the article, and approved the final draft.
- Yi Sun performed the experiments, prepared figures and/or tables, and approved the final draft.
- Huacai Zhao conceived and designed the experiments, analyzed the data, prepared figures and/or tables, authored or reviewed drafts of the article, and approved the final draft.
- Wei Ren performed the experiments, prepared figures and/or tables, and approved the final draft.
- Dan Gao conceived and designed the experiments, authored or reviewed drafts of the article, and approved the final draft.
- Zhigang Wang performed the experiments, analyzed the data, authored or reviewed drafts of the article, and approved the final draft.
- Wei Lv conceived and designed the experiments, prepared figures and/or tables, and approved the final draft.

- Qingchuan Dong performed the experiments, analyzed the data, authored or reviewed drafts of the article, and approved the final draft.

## Human Ethics

The following information was supplied relating to ethical approvals (i.e., approving body and any reference numbers):

All samples obtained in this study were approved by the ethics committee of Shaanxi Provincial People's Hospital and abided by the ethical guidelines of the Declaration of Helsinki.

## Animal Ethics

The following information was supplied relating to ethical approvals (i.e., approving body and any reference numbers):

Animal experiments were approved and supervised by the Animal Ethics Committee of Shaanxi Provincial People's Hospital.

## Data Availability

The raw data is available in the Supplementary Files.

## Supplemental Information

Supplemental information for this article can be found online at http://dx.doi.org/10.7717/peerj.16314#supplemental-information.

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
