# Peer review of "Niraparib restrains prostate cancer cell proliferation and metastasis and tumor growth in mice by regulating the lncRNA MEG3/miR-181-5p/GATA6 pathway"

_PeerJ, doi:10.7717/peerj.16314_

## Round 0.1 · original submission · Major Revisions

· Academic Editor

Major Revisions

Please carefully read the comments and suggestions from Reviewers and provide your point-by-point responses.

Reviewer 1 ·

Basic reporting

The manuscript presents an interesting investigation into the molecular mechanisms of niraparib in prostate cancer, focusing on the MEG3/miR-181-5p/GATA6 pathway. It makes a significant contribution to the field by filling a knowledge gap and proposing a novel molecular mechanism of action for niraparib. The technical standard employed is generally high, but additional information is needed to verify the availability, robustness, statistical soundness, and control measures of the underlying data. Overall, the manuscript has the potential to enhance our understanding of PCa progression and provide a foundation for future research.

Experimental design

1. Provide more details about the inclusion and exclusion criteria for the PCa patients recruited.
2. Specify the method used for collecting PCa tissues and non-tumor adjacent tissues during the survey. Include information about the storage conditions and handling of the samples.
3. Provide more information about the construction of the MEG3 and GATA6 reporter vectors, such as the cloning strategy and the controls used to validate the vectors.
4. Specify the imaging technique used to capture images of the excised tumors. Additionally, describe the method used for tumor volume measurement and the unit of measurement used
5. Provide more details about the co-expression network analysis mentioned in section 3.1. Explain the criteria used to establish the network and how the MEG3/miR-181-5p/GATA6 axis was determined from this analysis.

Validity of the findings

1. Explain the rationale behind choosing niraparib as a potential treatment for PCa in section 3.2. Provide background information on the mechanism of action of niraparib and how it relates to the MEG3/miR-181-5p/GATA6 axis.
2. Explain the rationale for using a dual-luciferase reporter assay in section 3.4. Describe how this assay was performed and how it confirms the interaction between miR-181-5p and MEG3/GATA6.
3. Include positive and negative controls for the gene expression modulation experiments mentioned in section 3.4. This will help validate the observed effects of MEG3 and miR-181-5p on GATA6 expression.

Additional comments

1. The introduction should provide more context on why PCa is considered a lethal cancer and why it is important to study it.
2. If possible, the reference to "cancer statistics of 2022 in the United States" should be updated to the most recent available data to ensure accuracy
3. Specify the role of PARPi in improving overall survival rate. Clarify how PARPi treatment, specifically olaparib, improves the overall survival rate in metastatic CRPC patients with homologous recombination repair defects. Discuss the response rates and evidence for PARPi in treating CRPC.
4. Emphasize the need to investigate whether PARPi exerts antitumor effects through regulating the transcriptome level. Explain the potential significance of clarifying this mechanism.
5. Clarify the link between MEG3 and DNA repair genes. Explain how MEG3 expression is related to DNA repair genes like PTEN and their effect on proliferation, metastasis, and apoptosis. Discuss the implications of MEG3 expression for PARP targeted CRPC therapies. Then, highlight the research gap regarding the effect of PRAPi on MEG3 expression
6. Instead of using the phrase "our study proposed the reasonable hypothesis," it would be more appropriate to say "Based on these findings, we hypothesized that..." or "We posited the hypothesis that..." This would strengthen the scientific language and tone of the manuscript.

Reviewer 2 ·

Basic reporting

Good

Experimental design

A. Mention the passage of time in the sentence "Cells were maintained in DMEM containing 10% fetal bovine serum and antibiotics at 37°C under 5% CO2 at 37°C." Correct the redundant "at 37°C" repetition.
B. Detail the reasons for using specific reference genes for normalization, such as GAPDH and U6. Explain why these genes are suitable for normalizing lncRNA MEG3 and GATA6 expression levels.
C. Specify the amount and concentration of primers used for RT-qPCR analysis. Ensure the specificity of the primers by mentioning any validation steps taken.
D. Explain how the binding sites of miR-181-5p in MEG3 and GATA6 were searched in the Starbase software. Provide the version of the software used and the specific criteria applied for identifying the binding sites.
E. Provide information about the positive and negative controls used in the immunohistochemistry assay.
F. Provide more details about the number of replicates used for each experiment or analysis.
G. Provide information on the source of PC3 cells and their characteristics (e.g., cell line authentication, passage number) to ensure reproducibility of the experiments.

Validity of the findings

The manuscript lacks information on the availability and robustness of the underlying data. Details about sample size, the reproducibility of experiments, and statistical analyses performed should be included to ensure transparency and soundness.

Additional comments

A. Consider mentioning the impact of PCa on mortality rates or its burden on society. Include the specific source or reference for the statistics mentioned.
B. Provide a more detailed explanation of how PARP inhibitors (PARPi) work and their impact on DNA damage repair and apoptosis. Besides, explain how synthetic lethality and homologous recombination repair gene defects are related to PARPi effectiveness.
C.Provide more background information on the role of lncRNAs in PCa development and their relevance to the topic.

Reviewer 3 ·

Basic reporting

1. Explain why it is relevant to mention the comparison with lung cancer and digestive cancers in introduction. What is the significance?
2. Specify what these limitations are and why they are significant. Provide some details on the rates of treatment failure or recurrence for these therapies in PCa patients.
3. Mention the limitations or challenges associated with the various new drugs developed for CRPC. Provide examples or references to support these limitations.
4. Provide more information on the effectiveness and clinical outcomes of PARPi in breast and ovarian cancer treatment. Explain how the application of PARPi has been expanded to treat advanced PCa and the specific benefits observed.
5. Provide a clearer transition between the introduction and the discussion of MEG3. It is not immediately clear why MEG3 is being discussed, so a logical connection should be established.
6. The sentence "Our subsequent study confirmed miR-181a-5p as a target of MEG3 in PCa cells" requires more explanation. How was this confirmation performed? Was it through experimental validation or through bioinformatic analysis? Include details about the methods used to confirm the relationship between MEG3 and miR-181a-5p.

Experimental design

1. Explain the rationale behind selecting the different concentrations of niraparib (0, 1, 2, 4, 8 μM) and the different incubation times (0, 30, 60, 120, 240 min) for treatment. Justify these choices based on previous studies or experimental considerations.
2. Provide additional information about the Lipofectamine 3000 transfection protocol. Include details about the concentration of the transfection reagent and the incubation time for efficient transfection.
3. Clarify the purpose of the CCK-8 assay and how it measures cell viability. Provide additional information on the incubation time and conditions for the CCK-8 assay.
4. Explain the protocol for the wound healing assay in more detail. Include the size and number of scratches made, culture conditions, and the distance measured for wound healing analysis.
5. Provide more details about the Transwell invasion assay such as the size of the Transwell chambers (pore size), the amount of Matrigel used for coating, and the duration of the invasion assay.
6. Specify the concentration of the primary antibodies used in the Western blot analysis. Additionally, provide information on the dilution factor of the primary antibodies and the concentration of the secondary antibody.
7. State and justify the choice of 4 nM niraparib concentration in section 3.3. Explain how this concentration was determined and why it was chosen for investigating the effects on PCa cell behaviors.
8. Provide a more detailed explanation of the search criteria and parameters used in the Starbase software mentioned in section 3.4. This will help readers understand how miR-181a-5p was predicted to have complementary binding sites with MEG3 and GATA6.

Validity of the findings

The methods employed in the study are described in a satisfactory manner, allowing for replication. However, additional details regarding the specific techniques, reagents, and equipment used would enhance the methodological section. This would ensure the replicability of the study by other researchers interested in further investigating the proposed molecular mechanisms.

---

## Round 0.2 · Minor Revisions

· Academic Editor

Minor Revisions

Please carefully read the comments and suggestions from the Section Editor below:

"As I began reading this work, certain concerns started to arise. While I haven't gone through the manuscript in detail, the primary issues appear to be related to language, grammar, verb tense usage, coherence in linking ideas, and punctuation, among others.

One notable area of concern is the materials and methods section, which appears to be lacking essential information. It's important to emphasize that revisions should be aimed at providing readers with a more comprehensive understanding of the experimental processes. Adjustments are necessary to ensure additional clarity and specificity in explaining the experimental procedures. Such clarifications are essential to guarantee that the methods and procedures can be easily understood by readers of a scientific paper, and importantly, can be reproduced.

Additionally, there is no description of the image analysis process in the manuscript. This is a crucial aspect that needs to be addressed.

I'd like to remind you to upload the images of the original blots, ensuring that the ladder is clearly visible. It's essential to present data that is clear, transparent, and upholds the highest standards of scientific integrity.

Furthermore, please consider highlighting the relevance of the work within the manuscript. Discuss the translational aspects or the motivating factors behind this research. It's important to convey what inspired you to undertake this study, especially given that similar research has been conducted and even the FDA has approved this drug for prostate cancer. (and more recently, "On August 11, 2023, the Food and Drug Administration approved the fixed dose combination of niraparib and abiraterone acetate (Akeega, Janssen Biotech, Inc.), with prednisone, for adult patients with deleterious or suspected deleterious BRCA-mutated castration-resistant prostate cancer (mCRPC)". Providing this context will help readers understand the significance and contribution of this work more effectively."

Please provide your point-by-point responses to the comments and submit a revised manuscript.

**Language Note:** The review process has identified that the English language must be improved. PeerJ can provide language editing services - please contact us at [email protected] for pricing (be sure to provide your manuscript number and title). Alternatively, you should make your own arrangements to improve the language quality and provide details in your response letter. – PeerJ Staff

Reviewer 1 ·

Basic reporting

no comment

Experimental design

no comment

Validity of the findings

no comment

Additional comments

The author has made good revisions to this article and carefully revised my opinions. I think this article has met the publishing standards.

Reviewer 2 ·

Basic reporting

No.

Experimental design

No.

Validity of the findings

No.

Additional comments

Your revised manuscript is acceptable for publication.

Reviewer 3 ·

Basic reporting

The authors have addressed all of my comments and have incorporated those in the revised manuscript. Hence, I recommend to accept it for publication.

Experimental design

I have no other review comments.

Validity of the findings

I have no other review comments.

Additional comments

I have no other review comments.

---

## Round 0.3 · accepted · Accept

· Academic Editor

Accept

The authors have addressed all comments from the Section editor.

eg lines 112-124: "Although numerous biomarkers, such as BRCA mutations and other genetic mutations related to HR, have been explored. however, there are still no gold standards"